# Increased Encapsulation Efficiency of Methotrexate in Liposomes for Rheumatoid Arthritis Therapy

**DOI:** 10.3390/biomedicines8120630

**Published:** 2020-12-18

**Authors:** Diana Guimarães, Jennifer Noro, Ana Loureiro, Franck Lager, Gilles Renault, Artur Cavaco-Paulo, Eugénia Nogueira

**Affiliations:** 1Centre of Biological Engineering, University of Minho, 4710-057 Braga, Portugal; dianapguimaraes@ceb.uminho.pt (D.G.); jennifer.noro@ceb.uminho.pt (J.N.); aloureiro@ceb.uminho.pt (A.L.); artur@deb.uminho.pt (A.C.-P.); 2INSERM—Institut National de la Santé et de la Recherche Médicale, U1016, Institut Cochin, 75014 Paris, France; franck.lager@inserm.fr (F.L.); gilles.renault@inserm.fr (G.R.); 3Solfarcos—Pharmaceutical and Cosmetic Solutions, 4710-053 Braga, Portugal

**Keywords:** liposomes, methotrexate, encapsulation, pre-concentration ethanol injection method, rheumatoid arthritis

## Abstract

Methotrexate (MTX) is a common drug used to treat rheumatoid arthritis. Due to the excessive side effects, encapsulation of MTX in liposomes is considered an effective delivery system, reducing drug toxicity, while maintaining its efficacy. The ethanol injection method is an interesting technique for liposome production, due to its simplicity, fast implementation, and reproducibility. However, this method occasionally requires the extrusion process, to obtain suitable size distribution, and achieve a low level of MTX encapsulation. Here, we develop a novel pre-concentration method, based on the principles of the ethanol injection, using an initial aqueous volume of 20% and 1:1 ratio of organic:aqueous phase (*v*/*v*). The liposomes obtained present small values of size and polydispersity index, without the extrusion process, and a higher MTX encapsulation (efficiency higher than 30%), suitable characteristics for in vivo application. The great potential of MTX to interact at the surface of the lipid bilayer was shown by nuclear magnetic resonance (NMR) studies, revealing mutual interactions between the drug and the main phospholipid via hydrogen bonding. In vivo experiments reveal that liposomes encapsulating MTX significantly increase the biological benefit in arthritic mice. This approach shows a significant advance in MTX therapeutic applications.

## 1. Introduction

Methotrexate (MTX) is an effective drug used to treat autoimmune and inflammatory diseases such as rheumatoid arthritis, Crohn’s disease, multiple sclerosis, and psoriasis [1,2]. However, free MTX has some limitations that restrict its use such as its poor bioavailability, low specificity, drug resistance, and dose-dependent side effects [3]. Innovative strategies have been investigated to increase the therapeutic effect of drugs [4,5,6]. Because of their lipid composition and structural similarity to cellular membranes, liposomes are considered the most used drug delivery system for the intracellular delivery of drugs [7]. Furthermore, they can encapsulate both hydrophilic and hydrophobic drugs [8]. A recent study of our research group reported a liposomal formulation encapsulating MTX that was shown to be a good therapeutic delivery system as demonstrated by its biological effect in vitro and in vivo [9,10].

The production of liposomes involved multiple steps in a complex and precise process that has a critical impact on the final liposome characteristics, such as size, stability, and functionality of the finished liposomes [11,12]. The more adequate method for liposome production and drug encapsulation also depends on the physicochemical characteristics of the drugs to be encapsulated [13]. The ethanol injection method is commonly used to produce liposomes, due to its simplicity, ease of scale-up, and safe production technique. Liposomes obtained by the ethanol injection method were spontaneously formed when the organic phase containing the dissolved lipids was rapidly injected into an aqueous phase by agitation [14]. This method does not induce oxidative and degradation modifications in most encapsulated drugs or in the lipid components [15,16]. However, to reduce liposome size and form unilamellar vesicles, extrusion should be applied [17].

The encapsulation efficiency (EE) of the drug is determined by the properties of the liposomes, including their aqueous volume or membrane rigidity. Furthermore, the encapsulation is affected by hydrophilic or hydrophobic properties of drugs and their capacity to interact with the membrane bilayer [18,19,20]. There are two different approaches for the encapsulation of drugs into liposomes. In the active method, usually remote loading, the drug is encapsulated in liposomes after their preparation, through a transmembrane gradient method, with an EE of around 100%. Since the MTX molecule is not considered a weak amphipathic acid or base, the remote loading method cannot be used [21]. MTX is encapsulated in liposomes by a passive method, that corresponds to the encapsulation of drugs during the liposome formation or in a phase of preparation when the liposomal structure is extremely fluid [22]. The passive encapsulation of hydrophilic drugs depends on the capacity of liposomes to trap the aqueous phase containing the drug. This methodology results in lower EE, since the drug retention is limited to the size of the aqueous compartment in liposomes and drug solubility [4].

The purpose of our study was to increase the encapsulation of MTX in liposomes, through the development of a novel production method based on the principles of ethanol injection. The use of a reduced initial aqueous volume (pre-concentration) and optimization of the organic:aqueous ratio proves to be essential to obtain a suitable size distribution and higher drug EE. The molecular interactions occurring between the MTX and the main phospholipid present in the liposomal bilayer were evaluated through nuclear magnetic resonance (NMR) studies. The biological benefit of the liposomes produced by the novel method was proved in a mouse model of arthritis.

## 2. Experimental Section

### 2.1. Materials

1,2-dioleoyl-sn-glycero-3-phosphoethanolamine (DOPE), egg phosphatidylcholine (EPC), and 1,2-distearoyl-sn-glycero-3-phosphoethanolamine-N-[methoxy(polyethylene glycol)-2000] (DSPE–mPEG) were obtained from Lipoid GmbH (Ludwigshafen, Germany). Deuterium oxide (D2O) and deuterated dimethyl sulfoxide (DMSO-d6) were acquired from Cortecnet (Les Ulis, France). All the other chemicals involved in this work were purchased from Sigma-Aldrich (St. Louis, MO, USA), except the MTX, which was acquired from Huzhou Zhanwang Pharmaceutical (Huzhou City, China), cholesterol from Anhui Chem-Bright Bioengineering Co Ltd. (Huaibei, China)*,* type II bovine collagen purchased from Chondrex, Morwell Diagnostics (Zumikon, Switzerland), and complete Freund’s adjuvant used in mouse experiments from Fisher Scientific (Ilkirch, France). All compounds were used without further purification.

### 2.2. Liposome Production by Ethanol Injection Method

Liposomes composed of DOPE or EPC/cholesterol/DSPE-mPEG [9] were produced by the ethanol injection method. Lipids were weighted at the initial molar ratio of 54:36:10. *MTX* disodium salt (soluble in aqueous buffer) was prepared by adding two NaOH molar equivalents to phosphate-buffered saline (PBS) containing commercial MTX. After the complete solubilization of MTX, the pH was adjusted to pH 7.4.

#### 2.2.1. Conventional Method

For the production of 10 mL of liposomes, DOPE, cholesterol, and DSPE-mPEG were dissolved in 2 mL of ethanol. The organic phase was added through gravity, using a 20G needle coupled to a plastic syringe, under magnetic stirring (500 rpm) to 10 mL of PBS (pH 7.4), at 70 °C. MTX disodium salt, as a hydrophilic drug, was added in the aqueous phase (PBS), to a final concentration of 20 mg/mL. The liposomes were then extruded (extruder supplied by Lipex Biomembranes Inc., Vancouver, BC, Canada) several times through polycarbonate filters of 200 nm and then 100 nm pore sizes (Nucleopore) to form unilamellar liposomes.

#### 2.2.2. Pre-Concentration Method

The lipid components were dissolved in different volumes of ethanol (1, 2, and 4 mL), to further obtain different initial ratios of organic:aqueous phase (*v*/*v*) in liposomes, of 1:2, 1:1, and 2:1, respectively. The organic phase was added through gravity, using a 20G needle coupled to a plastic syringe under vigorous magnetic stirring (500 rpm) to 2 mL of PBS (pH 7.4) containing 100 mg/mL of MTX, at 70 °C. After ethanol evaporation, the remaining 8 mL of PBS was added to achieve the final volume (10 mL) and MTX concentration (20 mg/mL). The liposome suspension was then kept under stirring for 15 min at room temperature. When necessary (size ≥ 150 nm and polydispersity index (PDI) > 0.1), liposomes were extruded several times through polycarbonate filters of 200 nm and then 100 nm pore sizes (Nucleopore). Where indicated, liposomes were prepared with D_2_O instead PBS buffer.

### 2.3. Determination of MTX Concentration

The non-encapsulated MTX was removed from the liposomes after passage through a gel filtration chromatography column (GE Healthcare, UK), with 5 kDa cut-off (PD-10 Desalting Columns containing 8.3 mL of Sephadex™ G-25 Medium). After separation, the MTX concentration was determined by measuring the absorbance at 303 nm, the maximum wavelength of MTX in PBS, a method previously validated [23]. Briefly, ultraviolet–visible spectra of liposome-encapsulated MTX were recorded on a BioTek Synergy™ HT spectrophotometer using a quartz microplate. The absorbance at 303 nm in empty liposomes was used as a blank. The final MTX concentration was determined based on the respective calibration curve. The encapsulation efficiency (EE) is defined by the concentration of the encapsulated MTX detected in the final liposomal formulation over the initial MTX concentration used to make the liposomal formulation. EE (%) was calculated using the following equation.
(1)$$\mathrm{EE}\;\left(\%\right)=\frac{\left[\mathrm{MTX}\right]\;\mathrm{encapsulated}\;\mathrm{in}\;\mathrm{liposomes}\;}{\left[\mathrm{MTX}\right]\;\mathrm{initial}\;\mathrm{used}\;\mathrm{in}\;\mathrm{liposomes}\;\mathrm{preparation}}\;\times \;100$$

### 2.4. Physicochemical Characterization of Liposomes

The physicochemical characterization of liposomes was evaluated in terms of size distribution using a dynamic light-scattering technique. The analysis was determined at pH 7.4 (PBS) and at 25.0 °C, using a Malvern Zetasizer Nano ZS, Malvern Instruments (Malvern, UK), with photon correlation spectroscopy. The viscosity and refractive index of dispersant were 0.8616 cP and 1.332, respectively. Each sample was measured in triplicate and the results are presented as mean value ± standard deviation (SD). The stability of liposomes over time was evaluated at 4 °C for 12 weeks, by monitoring changes in liposome size and PDI.

### 2.5. Evaluation of Compound Interactions by ^1^H NMR

^1^H NMR experiments were performed using a Bruker Avance III Instrument, operating at 400 MHz. To evaluate the distribution of MTX through the liposomes’ bilayer, the liposomes were formulated directly in D_2_O. The procedure was followed as mentioned in Section 2.2.2 but involved replacing the aqueous phase PBS buffer with D_2_O. To evaluate the lipid–compound interactions, DOPE or EPC was mixed in DMSO-d_6_ with an equimolar ratio of MTX or *N*-protected aspartic acid ((*S*)-2-(3-(naphthalen-1-yl)thioureido)succinic acid).

### 2.6. Synthesis of (S)-2-(3-(Naphthalen-1-yl)Thioureido)Succinic Acid

To a round-bottomed flask with L-aspartic acid (**A**, 0.13 g, 0.98 mmol) and 1-naphthyl isothiocyanate (**B**, 0.18 mg, 0.98 mmol), 2 mL of pyridine were added (60 % in water). The suspension was placed in an oil bath at 40 °C and kept stirred for 24 h. The solvent was then removed in a rotary evaporator. *N*-amino-protected L-aspartic acid (**C**) was obtained as a pure white solid after recrystallization with ethyl acetate (0.12 g, 0.38 mmol, η = 39 %). ESI (*m/z* = 329.18). NMR (DMSO-d_6_) δ_H_: 2.42 (dd, *J* = 15.6, 2.8 Hz, 1H); 2.72 (dd, *J*= 16.0, 11.2 Hz, 1H); 3.78 (dd, *J* = 11.2, 2.4 Hz, 1H); 7.49–7.62 (m, 4H); 7.84 (d, *J* = 8.0 Hz, 1H); 7.96 (d, *J* = 7.6 Hz, 1H); 8.01 (d, *J* = 8.4 Hz, 1H); 9.85 (s, 1H) ppm (Scheme 1).

### 2.7. Collagen-Induced Arthritis (CIA)

Six-week-old male DBA/1 mice, which are susceptible to CIA, were purchased from Janvier Laboratory (Le Genest-St-Isle, France). Mice were housed in groups of 6 per cage. Arthritis was induced with type II bovine collagen (CII). Male DBA/1 mice were injected subcutaneously at the base of the tail with 10 mg of CII emulsified in complete Freund’s adjuvant. On day 21, mice were boosted with a subcutaneous injection of CII in incomplete Freund’s adjuvant. In this model, arthritis develops 20 to 30 days after the first collagen injection [24]. Mice were monitored for evidence of arthritis in 4 paws using a blind procedure by a trained operator for arthritis scoring (20 years of experience). The severity of arthritis was evaluated using a clinical scoring front (4 fingers average, tarsus) and hind paws joints (5 fingers average, tarsus and ankle). Each joint was given a score ranging from 0 to 4 (0: normal joint, 1: erythema, 2: swelling, 3: deformity, 4: ankylosis) and summed, leading to a mouse individual score ranging from 0 to 40. As this model is strongly cage dependent, groups were carefully randomized among mouse cages. Treatments started on day 14, i.e., one week before the immune boost on day 21, and continued throughout the testing period. All animals received the same injection volume for different treatments, intraperitoneally (IP), twice a week, of free MTX at 7 mg/Kg and liposome-encapsulated MTX at 2 mg/Kg (according to the preliminary study). Mice were scored on the same day. This study was approved by the local animal experimentation ethics committee (CEEA34, Comité d’ethique en experimentation animal de l’Université Paris Descartes) under agreement no. APAFIS#9696-2017031016246908 (1 December 2017). All experiments were carried out in accordance with the National Institutes of Health guide for the care and use of laboratory animals (NIH Publications No. 8023, revised 1978).

Average severity scores per group and average weight variations per group were analyzed. The area under the curve (AUC) of severity curves was calculated in Excel. A therapeutic index was calculated on day 36 using the following equation:(2)$$Therapeutic\;Index\;\left(\%\right)=\frac{\left(AUC_{v}-AUC_{p}\right)}{AUC_{v}}$$
where *AUC_p_* stands for area under the curve of the given product tested and *AUC_v_* stands for the area under the curve of the vehicle group (PBS). This index somewhat reflects the percentage of the reduction of the severity of arthritis.

### 2.8. Statistical Analysis

A non-parametric one-way analysis of variance Kruskal–Wallis test on day 36 was used to test for significant differences between groups. The Dunn post hoc test without correction was then used to identify differences between all groups. All analyses were performed using R software and the “dunn.test” library for Dunn post hoc tests.

## 3. Results

### 3.1. Liposome-Encapsulated MTX Prepared by the Conventional Ethanol Injection Method

The liposomal formulation used in this study (DOPE:cholesterol:DSPE-mPEG, 54:36:10 molar ratio) was previously proved to be a good therapeutic delivery system of MTX, as demonstrated by the biological effect in vitro as well in vivo [9,10].

Liposome-encapsulated MTX produced by the conventional ethanol injection method (Liposome A) presents a high value of size and PDI (Table 1). These values are considered not suitable for further in vivo applications (size < 150 nm and PDI ≤ 0.1), as the extrusion process is crucial to achieve liposomes with an appropriate size distribution. Furthermore, the low EE (%) of MTX decreases after the extrusion process (from 1.33 ± 0.19 to 0.88 ± 0.16).

### 3.2. Liposome-Encapsulated MTX Prepared by the Pre-Concentration Ethanol Injection Method

The influence of three different initial ratios of organic:aqueous phase (*v*/*v*), 1:2, 1:1, and 2:1 (Liposomes B, C, and D, respectively), was evaluated, initially using 20% of aqueous volume (pre-concentration) and adding the remaining 80% at the end of the process. The results showed that this method promotes an increase in EE (%) for all liposomes (Table 1). However, high values of size and PDI are obtained when ratios 1:2 and 2:1 (Liposomes B and D, respectively) are used. The extrusion of these liposomal formulations is imperative to achieve suitable size distribution but leads to a decrease in methotrexate concentration. Only the ratio 1:1 (organic:aqueous phase, *v*/*v*) achieved suitable values of size (128.76 ± 7.78 nm) and PDI (0.107 ± 0.02) and a high EE level (22.90 ± 0.17%). In this way, Liposome C has similar physicochemical characteristics to the liposomes obtained by the conventional ethanol injection method after extrusion with the advantage of a higher MTX encapsulation. Liposome C was shown to be very stable through time, maintaining its size (127.60 ± 9.95 nm) and PDI (0.102 ± 0.02) without significant drug leakage, for at least 12 weeks. Figure 1 provides an overview of liposome production by both methods.

### 3.3. H NMR of Liposome-Encapsulated MTX

In order to study the effect of the extrusion process on the decrease in the EE, the ^1^H NMR approach was applied. Liposome-encapsulated MTX was directly prepared in deuterium oxide, as described in Section 2.2.2, and presented similar physicochemical characteristics to those prepared with PBS (data not shown). It should be noted that the NMR experiment was performed immediately after the removal of the non-encapsulated (free) MTX from the liposomes. In Figure 2 is depicted the ^1^H NMR of the aromatic area of the encapsulated MTX, (A) before and (B) after extrusion. The singlet signal corresponds to the proton of the pteridine ring, and both doublets to the phenyl ring. It is possible to observed that MTX is distributed between two different environments, one in which the signal is well defined (dashed line), and the other where the signals appear as broad singlets (solid line).

Given the two different environments observed for the MTX peaks, we may presume that the drug present near the surface of the liposome is represented by the well-resolved signals, since it should be more easily detected by the NMR apparatus. The drug present in the inner parts of the liposomes appears as broad singlets, given the presence of the lipid’s bilayer. The presence of lipids in NMR spectra is commonly associated with the loss of resolution of the proton peaks of the existent solution [25].

By the peak’s integration, it is possible to observe that 20% of the MTX is at the surface, while 80% is inside the liposomes. After extrusion, a pronounced decrease in the signal intensity is also observed, especially in the signals corresponding to the MTX at the liposome’s surface (dashed line).

### 3.4. Role of the Lipid in the Encapsulation of MTX

The lipids used for the liposome formation greatly affected the EE [26]. Therefore, liposomes composed with EPC (Liposomes E, Table 1) instead of DOPE were produced to observe its effect on MTX encapsulation.

From the results (Table 1), we may observe that the lipid used induced differences in the liposome size and EE. Regarding the differences detected in the EE (%) for liposomes composed of DOPE and EPC (22.90 ± 0.17, 11.85 ± 0.23, respectively), it is possible to observe that the drug could interact strongly with DOPE, leading to its higher retention in the formulation. Given the hydrophilic head of both lipids, EPC and DOPE, we proposed the possibilities of non-covalent interactions, as depicted in Figure 3. Nonetheless, we may not disregard other interactions that can occur between the two molecules, given the different chemical groups present.

To observe the interactions between the lipids with MTX, we used ^1^H NMR studies. ^1^H NMR spectra of the aromatic area (Figure 4) show that the addition of EPC to MTX (B) does not lead to any changes in the chemical shift of all represented peaks. Meanwhile, with DOPE (C), a chemical shift of the amide proton is observed from δ_H_ 8.07 to 7.88 ppm. The same behavior is observed for all peaks located in the glutamic moiety of MTX (data not shown).

### 3.5. Interaction of DOPE with N-Protected L-Aspartic Acid

In order to verify that the lipid–MTX interaction is based on the terminal amino-acid moiety of MTX, we used ^1^H NMR studies between DOPE and L-aspartic acid. The amino acid was *N*-protected with an aromatic group (naphthyl) to mimic the terminal structure of MTX (compound C in Scheme 1).

Figure 5 shows the ^1^H NMR spectra of the protected amino acid (A) and the protected amino acid with an equimolar amount of DOPE (B). It is possible to observe that in the presence of the lipid, the aliphatic signals of the aspartic acid lose resolution and/or suffer a change in their chemical shift. The same behavior was observed previously for MTX with DOPE, but not for MTX with EPC (Figure 4).

### 3.6. Biological Effect of Liposome-Encapsulated MTX

The influence of the preparation method in the biological effect was evaluated in vivo using a mouse model of arthritis (collagen-induced arthritis in the DBA/1 mouse strain). Liposome-encapsulated MTX was administered twice a week IP in arthritic mice, before disease onset. Liposomal formulations produced by both methods were analyzed in two independent experiments, with a control of free MTX being used for each one. The results showed that the injection of MTX in a soluble form has an impact on the prevention of arthritis development (Figure 6). However, the encapsulation of MTX in liposomes improved the prophylactic efficacy. Comparing the liposomes produced by both methods, we can observe that liposomes produced by the pre-concentration method demonstrated a similar biological benefit to the conventional method.

## 4. Discussion

Currently, MTX is considered the first line medication for rheumatoid arthritis patients [27]. Due to the excessive side effects, the encapsulation of MTX in liposomes can be an effective delivery system, reducing drug toxicity, while maintaining its efficacy. Ethanol injection is a common method to produce liposomes. It is known to be a simple, safe, reproducible, rapid, and easy to scale-up technique. However, this method occasionally requires the extrusion process, to obtain suitable size distribution, and achieve a low level of MTX encapsulation. The size, PDI, and EE of liposomes produced by the ethanol injection method can depend on several conditions, such as lipid solution injection rate, temperature, homogenization intensity, lipid concentration, and composition [28]. It is also known that the size of liposomes can be controlled by the ratio of ethanol to aqueous phase [16,29].

Based on the principles of the ethanol injection method (conventional method), we developed a novel strategy (pre-concentration method) to increase the EE of MTX in liposomes and to obtain small values of size and PDI (Figure 1). During both processes, an organic phase, composed of lipids dissolved in a ethanolic solution, is rapidly injected into an aqueous phase with vigorous magnetic agitation, promoting liposome formation [14]. The main difference is in the initial aqueous volume and the ratio of organic:aqueous phase (Figure 1). In the conventional method, we used an initial aqueous volume of 100% and a 1:5 organic:aqueous ratio (*v*/*v*), achieving a very low encapsulation of MTX (Table 1). These results are in agreement with the literature, where the encapsulation of most drugs by passive loading results in an EE around 1%, causing a large amount of drug waste [30]. Hydrophilic drugs such as MTX normally have a poor EE due to rapid migration and consequently the loss of drug into aqueous phase [31]. Indeed, due to the larger volume of the aqueous phase in the outside environment of the liposomes, compared to the limited aqueous volume inside the liposomes, the encapsulation of MTX results in low efficiency [32]. ^1^H NMR studies, used to evaluate the effect of extrusion on the decrease in the EE, show that the MTX at the liposomes’ surface also decreases after extrusion (Figure 2). The pressure applied during this procedure may cause a higher release of this MTX, leading to lower EE.

The use of an initial aqueous volume of 20% (pre-concentration) and a 1:1 organic:aqueous ratio (*v*/*v*) allows us to obtain liposomes with suitable size distribution and higher MTX encapsulation (Table 1, liposome C). The changes in the ratio of organic:aqueous phase determine the initial lipid and MTX concentration in the suspension and consequently influence the final physicochemical characteristics of liposomes. Through this pre-concentration method, we reduce the aqueous volume outside the liposomes, promoting the interaction between MTX and lipids.

The main lipids used to prepare liposomes are phospholipids. Their amphiphilic nature determines their precise distribution in the liposomal membrane. The carbon chains of phospholipids are considered hydrophobic regions and are aligned inside the lipid membrane. The polar heads of phospholipids are hydrophilic and are positioned on the inner and outer parts of the bilayer [33]. We compare liposomes produced with two different main lipids, DOPE and EPC, in order to evaluate their effect on MTX encapsulation (Table 1). Both lipids have similar aliphatic chains, however, they differ in their hydrophilic heads. EPC is composed of a quaternary amine, while DOPE is composed of a primary amine. Moreover, the phosphate group of EPC is ethylated, while in DOPE it remains negatively charged since it is not alkylated. These structural changes may induce a bulkier hydrophilic head group of EPC than DOPE, possibly leading to differences in packing during lipid bilayer formation, and consequently inducing discrepancies in the size achieved (Table 1). Considering the differences between the amine groups of both lipids, different possibilities of interactions are expected (Figure 3). Since DOPE is constituted by a primary amine, more possible interactions can occur (ionic and hydrogen bonds). In the case of EPC only, an electrostatic interaction can be established. The encapsulated drug can also affect the size of the formulation, given that the types of interaction that a drug can perform with a lipid membrane can modulate the proprieties of the lipid membrane [34].

NMR experiments performed to explore the mutual interactions between MTX and lipids suggest that an interaction between MTX and DOPE occurs, while the same interaction does not happen when EPC is used (Figure 4). This non-covalent bond could explain the increase in the EE when DOPE is used as the main lipid source, compared with EPC. Furthermore, the interaction between DOPE and *N*-protected L-aspartic acid suggests that the drug–lipid interaction is based on the terminal amino-acid moiety of MTX and occurs via hydrogen bonds (Figure 5). Based on NMR results, we hypothesized this possible interaction to justify the increase in EE in liposomes produced by the pre-concentration method. Although this interaction is the most plausible, we cannot exclude the possibility of other interactions occurring between the key molecules.

To prove the ability of our liposomes as a good delivery system, an in vivo assay was performed in an arthritis mouse model. The results show that the liposome-encapsulated MTX produced by this novel method maintains the prophylactic effect in the development of arthritis, presenting a similar biological benefit to the conventional method (Figure 6).

In conclusion, we have successfully developed a novel pre-concentration ethanol injection method to obtain higher MTX encapsulation in liposomes. The optimized method represents an appropriate alternative to the conventional ethanol injection method, avoiding the extrusion process for size reduction and enabling a greater increase in the EE and concentration of MTX. Based on our findings, we can suppose that the specific MTX–DOPE interaction occurs via hydrogen bonds, increasing the EE. These results suggest that the interaction of the highly charged drugs with liposomal membranes can be the driver for increased encapsulation in a novel liposome production method.

Furthermore, liposome-encapsulated MTX produced by the novel method, as well as the conventional method, significantly increases the biological benefit in an arthritis animal model. In this way, this approach is shown to be a significant advance in rheumatoid arthritis therapy using the liposomal encapsulation of MTX.

## 5. Patents

E.N., A.C.P., and D.G. filed for a patent to use the developed pre-concentration ethanol injection method for liposome production.

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
