# Peer review of "Increased Encapsulation Efficiency of Methotrexate in Liposomes for Rheumatoid Arthritis Therapy"

_biomedicines, 2020, doi:10.3390/biomedicines8120630_

Round 1

Reviewer 1 Report

Considering that the Authors responded to all issues raised by Editor and Reviewers, the manuscript is now suitable for publication.

Author Response

Reviewer 1

Comment:

“Considering that the Authors responded to all issues raised by Editor and Reviewers, the manuscript is now suitable for publication.”

Answer to comment:

We appreciate the opportunity to publish the manuscript in Biomedicines.

Reviewer 2 Report

The article by Nogueira et al. introduces an efficient way for drug loading using a method of pre-concentration ethanol injection. The description of the method is clearly presented, the possible reasons behind the higher EE are also suggested, and finally the biological studies of the drug loaded liposomes were performed.

I recommend the manuscript for acceptance and publication in Biomedicines after revisions.

I have a few comments to be considered:

Line 3/40 – As I understand, the liposomes were prepared using the combination of D2O and DMSO-d6, why not to use methanol-D4, which could be closer to the original method using ethanol (considering that the deuterated ethanol is rather expensive)?

Line 5/42, Table 1: Why not to convert the ratios into lowest possible numbers, i.e. 1:5, 1:1, and 2:1?  

Line 6/2 – Why the data are not shown? Should this be in supporting information or cited? Or the sentence fully avoided? Line 4 on this very page has the same comment.

7/3-11 In my opinion, three different forms of the guest molecule can be present in the sample. Either the molecule is present inside the liposome, or inside the lipophilic membrane, or there is a leakage of the drug from the liposome and it is partly present in the solution. I would suggest to reconsider the current conclusions and consider more the possibility of the free drug in solution for the well-resolved signals, which would be more in agreement with the general understanding and knowledge.

7/24-28 Figure 4 suggests that the amidic -NH- proton interacts through hydrogen bonding (HB). Nevertheless, would it be expected to have downfield chemical shift (deshielding effect) when being hydrogen bound? If the -NH- group is responsible for HB, why there are different interactions suggested in Figure 3? As the molecules, both MTX and the lipids, are rather complex, there is many possible HB functionalities (carbonyls, phosphate oxygen atoms, heterocyclic N, amino groups of MTX) which can play a role in HB. In my opinion it could be better to generalize the statement on their HB interaction rather than to try to point out specific interactions. In Figure 4 I can also recognize minor signals in A and B which are probably also suggesting that MTX is present in two different forms (similarly the aromatic signals). Figure 4, what is the chemical shift of proton on the chiral center of MTX which should be around 3.7 ppm?

8/7 It should be referred to the structure of the compound C in Scheme 1.

8/10 suffers instead surfers?

10/14 safe instead safety?

11/38 As noted above, there is many possible interactions which would have the character to participate in HB (carbonyls, phosphate oxygen atoms, heterocyclic N, amino groups of MTX), this is very difficult to predict without more thorough studies, which is not and maybe better if it wouldn’t be part of this manuscript.

11/51 Is than correctly used here? Similar sentence is on 9/13.

Author Response

Reviewer 2

“The article by Nogueira et al. introduces an efficient way for drug loading using a method of pre-concentration ethanol injection. The description of the method is clearly presented, the possible reasons behind the higher EE are also suggested, and finally the biological studies of the drug loaded liposomes were performed. I recommend the manuscript for acceptance and publication in Biomedicines after revisions. I have a few comments to be considered:”

Comment:

"Line 3/40 – As I understand, the liposomes were prepared using the combination of D2O and DMSO-d6, why not to use methanol-D4, which could be closer to the original method using ethanol (considering that the deuterated ethanol is rather expensive)?"

Answer to comment:

This point was clarified in the manuscript (Page 3, lines 38-41). To remark, to evaluate the distribution of MTX through the liposome’s bilayer, liposomes were prepared in D2O instead PBS buffer as aqueous phase. In another experiment, to evaluate the lipid-compound interactions, the DOPE or EPC was mixed in DMSO-d6 with MTX or N-protected aspartic acid.

Comment:

"Line 5/42, Table 1: Why not to convert the ratios into lowest possible numbers, i.e. 1:5, 1:1, and 2:1?"

Answer to comment:

The ratios were changed in Table 1 and throughout the manuscript, according to the reviewer’s comment.

Comment:

"Line 6/2 – Why the data are not shown? Should this be in supporting information or cited? Or the sentence fully avoided? Line 4 on this very page has the same comment."

Answer to comment:

The first sentence was deleted and the information relative to the second sentence was included in the manuscript (Page 6, line 3).

Comment:

"7/3-11 In my opinion, three different forms of the guest molecule can be present in the sample. Either the molecule is present inside the liposome, or inside the lipophilic membrane, or there is a leakage of the drug from the liposome and it is partly present in the solution. I would suggest to reconsider the current conclusions and consider more the possibility of the free drug in solution for the well-resolved signals, which would be more in agreement with the general understanding and knowledge."

Answer to comment:

This point was clarified in the manuscript (Page 6, lines 12-13).

Comment:

"7/24-28 Figure 4 suggests that the amidic -NH- proton interacts through hydrogen bonding (HB). Nevertheless, would it be expected to have downfield chemical shift (deshielding effect) when being hydrogen bound? If the -NH- group is responsible for HB, why there are different interactions suggested in Figure 3? As the molecules, both MTX and the lipids, are rather complex, there is many possible HB functionalities (carbonyls, phosphate oxygen atoms, heterocyclic N, amino groups of MTX) which can play a role in HB. In my opinion it could be better to generalize the statement on their HB interaction rather than to try to point out specific interactions. In Figure 4 I can also recognize minor signals in A and B which are probably also suggesting that MTX is present in two different forms (similarly the aromatic signals). Figure 4, what is the chemical shift of proton on the chiral center of MTX which should be around 3.7 ppm?"

Answer to comment:

We agree with the reviewer’s comments and we may not disregard the possibility of other interactions between both molecules. This point was clarified in the manuscript (Page 7, lines 21-22). The chemical shift of proton on the chiral center of MTX is 3.9 ppm and after the interaction with DOPE is 3.7 ppm.

Comment:

"8/7 It should be referred to the structure of the compound C in Scheme 1."

Answer to comment:

The information was included in the manuscript (Page 8, lines 7-8).

Comment:

"8/10 suffers instead surfers?"

Answer to comment:

This word was modified in the manuscript (page 8, line 11).

Comment:

"10/14 safe instead safety?"

Answer to comment:

This word was modified in the manuscript (page 10, line 14).

Comment:

"11/38 As noted above, there is many possible interactions which would have the character to participate in HB (carbonyls, phosphate oxygen atoms, heterocyclic N, amino groups of MTX), this is very difficult to predict without more thorough studies, which is not and maybe better if it wouldn’t be part of this manuscript."

Answer to comment:

This point was revised in the manuscript (Page 11, lines 47-50).

Comment:

"11/51 Is than correctly used here? Similar sentence is on 9/13."

Answer to comment:

This word was revised in the manuscript (page 12, line 2; page 9, line 13)

Round 2

Reviewer 2 Report

My comments and questions have been aswered and I am happy to recommend the manuscript for its publication without any further delays. 

This manuscript is a resubmission of an earlier submission. The following is a list of the peer review reports and author responses from that submission.

Round 1

Reviewer 1 Report

The manuscript “Increased methotrexate encapsulation in liposomes for rheumatoid arthritis therapy” is an interesting work, but I think that it has serious underlying problem.

Major

The strategy adopted by Authors to insert the drug inside the liposomes, pre-concentration method, seems much more efficient than the conventional method, however when analyzed in animal model, both conventional and pre-concentrated methods produce liposomes showing exactly the same effectiveness. So considering that the major part of the manuscript regards the production and the analyses of liposomes, to my opinion it would be much more suitable to a journal with a more chemico-physical connotation. Moreover, the manuscript has been submitted to Immunology and Immunotherapy, which seems inappropriate.

Also the title suggests that those liposomes, containing a higher amount of methotrexate than “conventional” liposomes, would be more effective in the treatment of RA, but the results obtained in the animal model do not support this conclusion. At least the title should be changed.

Minor

It is not described the type of pre-concentrated liposomes has been administered to mice. B, C or D?

The figure 6C and the legend of figure 6 should be improved.

Author Response

Comment:

“1. The strategy adopted by Authors to insert the drug inside the liposomes, pre-concentration method, seems much more efficient than the conventional method, however when analyzed in animal model, both conventional and pre-concentrated methods produce liposomes showing exactly the same effectiveness. So considering that the major part of the manuscript regards the production and the analyses of liposomes, to my opinion it would be much more suitable to a journal with a more chemico-physical connotation. Moreover, the manuscript has been submitted to Immunology and Immunotherapy, which seems inappropriate.

Also the title suggests that those liposomes, containing a higher amount of methotrexate than “conventional” liposomes, would be more effective in the treatment of RA, but the results obtained in the animal model do not support this conclusion. At least the title should be changed.”

Answer to comment:

We agree that the submission of the manuscript to the section Immunology and Immunotherapy seems inappropriate. We think that some issue occurred in the process of manuscript submission.  If it was possible, we would like to change the section of the article to Biomedical Materials and Nanomedicine.

We believe that the title does not suggest that the liposomes prepared with the new method are more effective in the treatment of RA. We decide to keep the title.

Comment:

“2. It is not described the type of pre-concentrated liposomes has been administered to mice. B, C or D?

The figure 6C and the legend of figure 6 should be improved.”

Answer to comment:

This point was clarified in the text (page 11, line 21) and in the legend of Figure 6 (page 10, line 5). The legend of Figure 6 was also improved (page 10, lines 5-8).  

Reviewer 2 Report

The major concern:

  1. Because there is not any description of the phospholipid measurement in the material and method section. How did the encapsulation ration be calculated?
  2. In figure 2, the authors demonstrated the NMR spectrum of liposomal MTX before and after extrusion. Could the authors describe more precisely which kind of preparation process and formulation of test samples?
  3. Please explain the difference between dark and gray bars in MTX group of Figure 6C.

Minor concern:

1.In figure 6, it is better to use the term liposomal MTX, which is more specific to represent the liposome-encapsulated drug, instead of “liposome+MTX.”

Author Response

Comment:

“1. Because there is not any description of the phospholipid measurement in the material and method section. How did the encapsulation ration be calculated?”

Answer to comment:

This point was clarified in the text (page 2, line 39).

Comment:

“2. In figure 2, the authors demonstrated the NMR spectrum of liposomal MTX before and after extrusion. Could the authors describe more precisely which kind of preparation process and formulation of test samples?”

Answer to comment:

This point was clarified in the text (page 3, line 12 and page 6, line 6).

Comment:

“3. Please explain the difference between dark and gray bars in MTX group of Figure 6C.”

Answer to comment:

This point was clarified in the legend of Figure 6 (page 10, lines 7-8).

Comment:

“In figure 6, it is better to use the term liposomal MTX, which is more specific to represent the liposome-encapsulated drug, instead of “liposome+MTX.”

Answer to comment:

The term “liposome+MTX” was replaced by “liposomal MTX” in Figure 6 (pages 9 and 10).

Round 2

Reviewer 1 Report

I still think that the title should be changed.

The last sentence of the Discussion should be changed too, I suggest:

...liposomes encapsulating MTX produced by the novel method, as well as conventional method, significantly increases the clinical benefit in an arthritis animal model.

Author Response

Reviewer: 1

Comment:

  1. I still think that the title should be changed.

Answer to comment:

The title was changed in the manuscript.

Comment:

  1. The last sentence of the Discussion should be changed too, I suggest:

...liposomes encapsulating MTX produced by the novel method, as well as conventional method, significantly increases the clinical benefit in an arthritis animal model.

Answer to comment:

The sentence was modified in the text (page 12, lines 9-10).

Reviewer 2 Report

Major concern

  1. No clear answer about the phospholipid measurement in the revised version.
  2. With lipid as the matrix, the analysis of MTX with the spectrophotometer is not the proper way. 
  3. As we know, the buffer system of the solution will affect the chemical behavior of phospholipids. Therefore, using D2O to replace the PBS in "Evaluation of compounds interactions by 1H NMR" will decrease results reliability. 
  4. In table 1, the EtOH final volume percentage seems not correct.
  5. The authors did not answer the question about the difference in the MTX group in figure 6.

Author Response

Reviewer: 2

Comment:                               

  1. “No clear answer about the phospholipid measurement in the revised version.”

Answer to comment:

This information was clarified in the manuscript (page 2, line 40).

Comment:

  1. With lipid as the matrix, the analysis of MTX with the spectrophotometer is not the proper way.” 

Answer to comment:

This point was clarified in the manuscript (page 3, lines 19-21).

Comment:

  1. “As we know, the buffer system of the solution will affect the chemical behavior of phospholipids. Therefore, using D2O to replace the PBS in "Evaluation of compounds interactions by 1H NMR" will decrease results reliability. 

Answer to comment:

We understand the reviewer's observation. However, 1H NMR is an approach and the liposomes produced in D2O have similar physicochemical characteristics to liposomes prepared in PBS buffer. This information was highlighted in the manuscript (page 6, lines 5-8).

Comment:                                      

  1. “In table 1, the EtOH final volume percentage seems not correct.

Answer to comment:

We confirm that the EtOH final volume percentage was correct.

Comment:

“5. The authors did not answer the question about the difference in the MTX group in figure 6.”

Answer to comment:

This information was clarified in the text (page 9, lines 7-9) and in the legend of figure 6 (page 10, lines 8-10). 

Round 3

Reviewer 2 Report

  1. “No clear answer about the phospholipid measurement in the revised version.” means on page 2 line 40 just described the initial molar ratio of lipids not including the method used for measurement after final products got from the process
  2. “In table 1, the EtOH final volume percentage seems not correct. For example, if the "Initial ratio
    Organic:Aqueous phase (v/v)" is 1:5 means EtOH is 1/6 of the final volume. 
  3.